# New Possibilities for Evaluating the Development of Age-Related Pathologies Using the Dynamical Network Biomarkers Theory

**DOI:** 10.3390/cells12182297

**Published:** 2023-09-17

**Authors:** Kazutaka Akagi, Keiichi Koizumi, Makoto Kadowaki, Isao Kitajima, Shigeru Saito

**Affiliations:** 1Research Center for Pre-Disease Science, University of Toyama, Toyama 930-8555, Japan; 2Division of Presymptomatic Disease, Institute of Natural Medicine, University of Toyama, Toyama 930-0194, Japan

**Keywords:** dynamical network biomarkers theory, Raman spectroscopy, aging, resilience

## Abstract

Aging is the slowest process in a living organism. During this process, mortality rate increases exponentially due to the accumulation of damage at the cellular level. Cellular senescence is a well-established hallmark of aging, as well as a promising target for preventing aging and age-related diseases. However, mapping the senescent cells in tissues is extremely challenging, as their low abundance, lack of specific markers, and variability arise from heterogeneity. Hence, methodologies for identifying or predicting the development of senescent cells are necessary for achieving healthy aging. A new wave of bioinformatic methodologies based on mathematics/physics theories have been proposed to be applied to aging biology, which is altering the way we approach our understand of aging. Here, we discuss the dynamical network biomarkers (DNB) theory, which allows for the prediction of state transition in complex systems such as living organisms, as well as usage of Raman spectroscopy that offers a non-invasive and label-free imaging, and provide a perspective on potential applications for the study of aging.

## 1. Introduction

Aging is characterized by a progressive loss of physiological integrity, which leads to tissue dysfunction and an increased vulnerability to death. Physiological integrity, (i.e., the intrinsic capacity or resilience of a body system) is proposed as a stronger predictor than the presence of morbidities of subsequent deteriorations in health status [1]. Younger individuals have a robust recovering capacity, termed “resilience”, against iterated insults and stress. Resilience declines with age, contributing to the emergence of diseases that manifest clinically [2]. This concept of age-related loss of resilience is correlated with the numerous discoveries and recent advances in the field of basic aging research. The hallmarks of aging, which include genomic instability, telomere attrition, epigenetic alterations, a loss of proteostasis, disabled macroautophagy, deregulated nutrient sensing, mitochondrial dysfunction, cellular senescence, stem cell exhaustion, altered intercellular communication, chronic inflammation, and dysbiosis, have been identified as key drivers of aging [3,4]. These hallmarks are tightly connected to chronic diseases, but not to driving aging independently; rather, they are highly intertwined, and understanding the interplay between them is critically important [5,6].

Among these hallmarks, research on cellular senescence has shown remarkable progress in understanding the mechanisms of aging, as well as drug development. Therefore, the prediction, identification, characterization, and pharmacological elimination of senescent cells have gained significant attention, not only by researchers in the aging field, but also by the pharmaceutical industries. Cellular senescence is defined as the irreversible arrest of cell proliferation that occurs when cells are exposed to a variety of stressors, such as genotoxic agents, nutrient deprivation, hypoxia, mitochondrial dysfunction, or oncogene activation [7,8]. Although there are several common markers of senescent cells, including the induction of the senescence-associated secretory phenotype (SASP), γ-H2AX nuclear foci, phosphorylated p53, the presence of cyclin-dependent-kinase (CDK) inhibitors (p16 and p21), and senescence-associated β-galactosidase (SA-βgal) activity, as well as an increased cell size, all of which are widely used and studied in research, the lack of universal or specific markers is a major limitation for the identification and targeting of senescent cells in vitro and in vivo [9]. Furthermore, the phenotype of cellular senescence is highly heterogeneous and dynamic, and thus it is difficult to distinguish between the non-senescence state and the senescence state. Emerging technologies in the fields of multi-omics, imaging, bioinformatics, and machine learning have enabled the more precise identification of senescent cells [10]. This technological revolution gives us novel insights into understanding cellular senescence and organismal aging, but there are still some limitations and challenges [10].

Given that the appearance of senescent cells is considered to be a cell fate transition from the proliferative state to the non-proliferative state, similar to the critical transitions that occur during cell differentiation and symptom onset, it can be detectable by the dynamical network biomarkers (DNB) theory, which detects early warning signals just before bifurcation points [11], such as “the pre-disease state” (discussed below). This review presents the usage of the DNB theory in biological samples with multiple time points. We discuss the potential applications of the DNB theory, not only for detecting senescent cells, but also for disease prediction to prevent age-related disorders and achieve our goal of healthy longevity.

## 2. Dynamical Network Biomarkers Theory

### 2.1. The Concept of DNB Theory

In view of the bifurcation theory of dynamical systems, when organisms have a high resilience in a homeostatic state (e.g., physiologically young or healthy), a body system is stable, with fast and small fluctuations in terms of the value of a physiological parameter or gene expression. During aging with a gradual loss of resilience, on the other hand, the system enters an allostatic state [12], which leads to a dynamic change in the parameter that shows slow and large fluctuations. Then, the system falls into a different state via bifurcation, which is also known as the “critical transition”, and the thresholds at which this transition occurs called “tipping points” [13,14]. This other state corresponds to the disease state or the state of allostatic overload (discussed below), indicating the breakdown of a physiological network (Figure 1A). This phenomenon is a consequence of the accumulation of undetectable elements, and once it occurs, it is often too late and non-reversible, like diabetes. Accordingly, methodology through which we can identify a sign of this critical transition or the state just before the critical transition, namely “the pre-disease state”, is needed for disease prediction. It is noteworthy that the concept of the pre-disease state was first proposed in the world’s oldest medical textbook, “Yellow Emperor’s Inner Canon”, in China more than 2200 years ago. In this textbook, the pre-disease state is termed “Weibing” in Chinese, which corresponds to “Mebyo”, meaning “not sick yet” in Japanese. We define Mebyo (pre-disease state) as a fluctuated state that can not be strictly categorized as being either healthy or sick.

These critical transitions are a sudden and large-scale change in state and occur in many complex systems, including ecosystems, the climate, financial markets, and microorganism populations [15,16,17,18,19]. The most important indicator of whether a system is getting close to the critical transition is a phenomenon known in dynamical systems theory as “critical slowing down” [13,20]. Critical slowing down exhibits increases in both fluctuation and autocorrelation, as well as a slow recovery from perturbation just before the critical transition. For example, ~34 million years ago, the Earth changed suddenly from a tropical state to a state with ice caps, a shift known as the greenhouse–icehouse transition. Tripati et al. reported evidence for synchronous deepening and subsequent oscillations in the calcite compensation depth in the tropical Pacific and South Atlantic oceans ~42 million years ago, with permanent deepening 34 million years ago [17,21], indicating a critical slowing down phenomenon. These signs are termed “early warning signals” [13,14]. To apply the concept of critical slowing down, which is well-known for one-dimensional systems, to high-dimensional complex systems with nonlinear networks such as body systems, Chen et al. extended the idea of critical slowing down and proposed the dynamical network biomarkers theory [11,22] (Figure 1B).

In general, gene expression is a fundamentally stochastic process, which stems noise from randomness in mRNA synthesis [23]. Stochastic gene expression results in a heterogeneous cell population that has beneficial effects in some contexts. In aging, however, noise in gene expression increases with age, and it may mask the expression changes in the genes that, in particular, lead to the critical transition towards the disease state. Notably, a data analysis using the DNB theory (namely a DNB analysis) can extract the cluster of the genes that shows large fluctuations with the strongest correlations. Thus, DNB analysis is applicable to noisy and nonlinear datasets, including multi-omics data. A mathematical formulation of the DNB theory was well-reviewed by Aihara et al. [22].

### 2.2. The Applications of DNB Theory

The DNB theory has been applied to various disease models, from cellular to organismal levels (Table 1). These include predictions of viral infection such as influenza A (H3N2) and COVID-19 [24,25,26], hepatocellular carcinoma metastasis [27], drug resistance in breast cancer [28], photodamage responses in skin [29], and adeno-to-squamous transdifferentiation in lung cancer [30].

Understanding cell fate transitions and controlling them is critically important, not only in the field of developmental biology, but also in aging. Emerging technologies such as single-cell RNA sequencing (scRNA-seq) combined with bioinformatics approaches have shown great progress. For example, a method called cell state transition assessment and regulation (cSTAR) uses a nonlinear dynamical model that distinguishes cell states and is able to predict a mechanistic network that controls cell state transitions, as well as manipulates the cell states [37]. From a scRNA-seq dataset of mouse hematopoietic stem cells, a DNB analysis was able to distinguish three different types of transcriptomic variations that corresponded to different cell fates [31]. Strikingly, Li et al. recently reported a module-based DNB (M-DNB) model used for cell fate determination during the differentiation process in human embryonic stem cells (hESCs) [32]. This M-DNB model, which transforms gene expression information into gene modules/networks based on a protein–protein interaction network, is optimized for scRNA-seq datasets, which often contain higher levels of transcript amplification noise and drop-out events. Using this model, they identified the key regulators of hESC differentiation as the M-DNB factors, including *FOS*, *HSF1*, *MYCN*, *TP53*, and *MYC*, with *FOS* being activated in the first tipping point and *MYC* in the second [32]. There are common biological features between ESCs and cancer stem cells (CSCs) [38]. Therefore, a DNB analysis may help in identifying the master regulators for the critical transition points during dynamical cellular processes, including cancer initiation, metastasis, and recurrence.

### 2.3. Cancer and Cellular Senescence

Cellular senescence is one of the first defense mechanisms against tumor promotion during carcinogenesis. This process is known as oncogene-induced senescence (OIS), which suppress the pro-proliferative effects of oncogenic stimuli by forcing cells to become senescent and prevent the expansion of pre-cancerous cells [39]. Senescence is also induced by chemotherapeutic drugs or ionizing radiation during cancer treatment, termed therapy-induced senescence (TIS) [40,41]. Generally, the mechanisms underlying TIS are connected to the DNA damage response (DDR) that leads to blocking tumor cell proliferation. However, it was found that some senescent tumor cells can escape from cell cycle arrest and acquire stemness properties with a highly aggressive growth potential, which contradicts the dogma of the irreversible arrest of the cell proliferation phenotype in senescent cells [40,41,42,43,44]. This phenomenon is possibly characterized as “survival at the brink”, which is associated with tumor recovery and cancer relapse [44]. During this process, it may display large fluctuations in gene expression with strong correlations, which can be detected using the DNB theory. Notably, Jackson et al. and Huna et al. reported that the dual and heterogeneous up-regulation of two opposing regulators, p21CIP1 for senescence/apoptosis and OCT4A for stemness in the topoisomerase II inhibitor, Etoposide, induced senescent embryonal carcinoma cells [45,46], suggesting early warning signals of a cell-fate transition towards tumor recovery.

Numerous studies have demonstrated that polyploidization occurs in many types of tumors [44,47]. These polyploid tumor cells are referred to as Polyploid Giant Cancer Cells (PGCCs) [48,49]. The mechanism leading to the formation of PGCCs may depend on endoreplication, mitotic slippage, cytokinesis failure, cell fusion, or cell cannibalism [48]. PGCCs can appear in response to anti-cancer therapies and are now considered to be involved in the immortality, invasion, origin, metastasis, and resistance of tumor cells to radiotherapy and chemotherapy [49]. Cancer cells can undergo senescence due to TIS, but it appears that senescent cancer cells might do more harm than good and lead to cancer recurrence, which is frequently connected to coupling cellular senescence with polyploidization/depolyploidization [48,50]. However, the detailed mechanisms of PGCC formation and function remain unclear. It is noteworthy that a few populations of senescent embryonal lung human fibroblasts (IMR90 cells) after 32–34 passages became polyploid and displayed both senescence markers, p21CIP1 and p16INK4A, and the stemness marker NANOG [38,51]. This observation suggests that the tumor-related ploidy cycle is associated with a transient state between senescence and stemness, which may be captured by a DNB analysis. Since the incidence of cancer is accelerating globally [52], the development of a novel and safer therapeutic approach for cancer is urgently needed. Hence, we believe that the DNB theory could contribute to understanding cancer biology.

## 3. Raman Spectroscopy

### 3.1. A General Background on Raman Spectroscopy

One of the major technical issues in mapping senescent cells in vitro and in vivo is sample preparation, because the preparation and preservation of cells or tissues can affect the final proportion of them that are sampled. Raman spectroscopy, on the other hand, offers a non-invasive and label-free method that allows for monitoring the progression of biological events in real-time manner. Raman spectroscopy is based on the interaction between light that is focused on a sample and the chemical bonds within the materials to be analyzed. The majority of the light on a sample is scattered at the same frequency as elastic light, called Rayleigh scattering. Compared to Rayleigh scattering, inelastic or Raman scattering is a rare and comparatively weak phenomenon. Depending on the direction of the energy shift caused by molecular vibrations (Raman shift), scattered electrons are either at lower (Stokes Raman) or higher (anti-Stokes Raman) energy levels [53]. Thus, Raman spectroscopy provides information associated with molecular vibrations, known as “molecular fingerprints”. However, understanding the biological significance of the signals from multi-component samples such as cells and tissues is extremely difficult, as Raman spectroscopy only obtains information from molecular vibrations. To overcome this problem, the establishment of a database and evaluation systems for Raman spectra related to various biological samples is required [54]. Raman spectroscopy, on the other hand, is widely used to analyze a lot of biological events, including cell death [55], cell differentiation [56,57], immune cell activation [58,59], cancer [60,61], and human diseases such as osteoarthritis and Hirschsprung disease [62,63,64]. Although Raman fingerprinting often provides reliable information that directly associated with the diagnostic and prognostic markers for diseases, further analytical strategies, including hardware technologies and methodologies in machine learning, are needed for its clinical usage [65].

### 3.2. Raman Spectroscopy and Cellular Senescence

It is noteworthy that Raman spectroscopy has been successfully used to distinguish senescent cells, as well as to investigate other molecular changes that occur at the cell and tissue levels during aging [53,66]. For example, senescence-associated Raman signatures were obtained from a culture of human umbilical cord mesenchymal stem cells in the context of replicative senescence [67]. Similarly, four different human fibroblast cell strains from fetal foreskin or lung tissues (BJ, IMR-90, MRC-5, and WI-38) were analyzed using Raman and infrared spectroscopy, and common molecular features in the proteins and lipids across the cell lines were found in the senescent cells [68]. The same group expanded their approach, and distinguished the senescent cells from the proliferative population in a human dermal fibroblast three-dimensional skin model [69]. Changes in the proteins and lipids in the senescent cells were confirmed by a recently developed method called normalized Raman imaging (NoRI), which converts stimulated Raman scattering images to absolute concentrations through the normalization of the undesirable intensity variation caused by sample light scattering and provides the local concentrations of proteins, lipids, and water from live or fixed tissue samples with a high spatial resolution [70]. NoRI identified a reduced cytoplasmic dry mass and protein concentrations in doxorubicin-induced senescent cells in MDCK cell and A7 cell lines compared to proliferating interphase cells [70,71]. They treated the MDCK cells with the genotoxic drug, doxorubicin (100 ng/mL), for 48 h or 5 days and performed NoRI imaging. The induction of cellular senescence was confirmed by the negligible level of EdU incorporation, as well as the high level of SA-βgal activity with an increased cell mass in the treated cells [70,71]. These observations support the notion that cytoplasmic dilution is associated with senescent cells [72]. NoRI also showed lipid accumulation in the senescent MDCK cells, but not in the senescent A7 cells, suggesting heterogeneity in the metabolic reprogramming of cellular senescence [70]. Raman imaging can be obtained from various biological samples, which are prepared with dishes, slides, tissue/liquid biopsy, and intraoperative pathology diagnoses [65]. In addition, the integration of a Raman fiber optic probe and endoscope enables real-time in vivo detection to help diagnose the extent and site of cancer development [54]. Although the detection of senescent cells, which are extremely rare populations, in vivo is still challenging, we speculate that Raman spectroscopy imaging will provide novel opportunities for understanding cellular senescence in vivo.

### 3.3. Raman Imaging and DNB Analysis

Despite Raman imaging and DNB analyses demonstrating strong power in investigating cell fate transitions, no research has been performed with a combination of these tools. Accordingly, we observed T cell differentiation at 0 (naïve T cells), 2, 6, 12, 24, and 48 h (fully activated T cell) using Raman spectroscopy, then performed a DNB analysis [33]. We successfully distinguished the transition state from naïve to activated T cells at 6 h after activation in terms of Raman shifts (namely DNB Raman shifts), which were accompanied by changes in the cellular materials. Given that T cells are known to be fully activated at 48 h after induction based on the expression level of CD69, a molecular biomarker of T cell activation [58], we were able to detect the signals of T cell activation at a very early time point. Therefore, we propose that obtaining DNB Raman shifts will help to predict the development of senescent cells. We speculate that the use of Raman spectroscopy combined with a DNB analysis can detect “pre-senescence cells”, in which cells do not have cellular senescence properties, but are getting close to the phase transition towards senescent cells. If we could detect “pre-senescence cells”, it will open up new possibilities for senotherapeutic approaches.

## 4. Senolytics and Senomorphics

The emerging therapeutic strategies for targeting senescent cells are called senotherapies, which include senolytics and senomorphics. Senolytics is the selective elimination of senescent cells by small molecules, while senomorphics is the inhibition of pathological SASPs to cause senostasis (senescent cells stay there but are less harmful) [73]. The majority of the senolytics identified to date promote the apoptosis of senescent cells by targeting the key enzymes involved in cellular pro-survival and anti-apoptotic mechanisms, such as SRC kinases, BCL-2 family proteins, HSP90, PI3K-AKT, p53-FOXO4, GLS1, and others (Table 2). Although numerous studies have demonstrated that senolytics ameliorate several age-related diseases and increase the health span of model organisms, undesirable side effects due to removing senescent cells have been reported [74,75].

Senomorphics, on the other hand, is considered to be a safer alternative to senolytics, as it suppresses the unwanted SASP expressions from senescent cells rather than directly removing them. Senomorphics can directly or indirectly attenuate the SASP of senescent cells by inhibiting mTOR, NF-κB, SIRT1, p38MAPK, JAK-STAT, and other signaling pathways (Table 3). The safety concern associated with senomorphics is the potential suppression of the growth-promoting functions induced by the SASP, similar to those seen in senolytics. In addition, continuous treatment is needed to maintain the suppression of the SASP in the case of senomorphics, while the intermittent administration of senolytics appears to be as effective as continuous treatment for attenuating senescent cell burden [86].

The side effects of senotherapies are due to the high heterogeneity in gene expression and the diverse origins of senescent cells, as well as their beneficial effects on tissue repair and regeneration [106,107]. An elimination of all senescent cells or a general inhibition of the SASP might cause the detrimental effects; thus, developing universal senotherapeutic drugs is extremely challenging. Further studies are needed to understand the manifestation of senescent cells for the successful development of senotherapeutic interventions. Notably, an M-DNB analysis identified the tipping points of hESC differentiation and found the master regulator genes, which commit to its cell fate determination [32]. Thus, we speculate that an M-DNB analysis may help in finding the master regulators of senescent cell development at the “pre-senescence state”, and these genes might have the potential to be novel senotherapeutic targets. If we could intervene in “pre-senescence cells”, it may be possible to reverse the pre-senescence to the healthy state, preventing senescent cell burden and chronic diseases, as well as delaying multimorbidity and increasing health span. We believe that Raman spectroscopy will also help to assess the effect of “pre-senescence intervention” (Figure 2).

## 5. DNB Analysis in Metabolism

### 5.1. Identification of DNB Genes

A DNB analysis is also possible to apply to metabolic diseases, including metabolic syndrome and type 2 diabetes [34,36]. We investigated a mouse model of metabolic syndrome, Tsumura Suzuki Obesity Diabetes (TSOD) mice, which are an inbred strain that spontaneously display metabolic syndrome phenotypes that correspond to phenotypes in humans [108]. The TSOD mice sequentially displayed phenotypes including obesity, hyperglycemia, dyslipidemia, hyperinsulinemia, and diabetes starting at around 12 weeks of age. They showed non-alcoholic steatohepatitis (NASH) at around 24 weeks of age and hepatocellular carcinoma at around 48 weeks of age. We performed a microarray analysis using white adipose tissue from the TSOD mice, collected at pre-symptomatic stages from 3 to 7 weeks of age. Analyzing this dataset using a DNB analysis, we found a group of collectively fluctuated genes with a significant correlation in strength (namely DNB genes) at 5 weeks of age, that we refer to as the pre-disease state in this mouse model [34]. In this study, we obtained 147 DNB genes, which are mostly associated with reproduction such as spermatogenesis and spermatid development. It is noteworthy that testosterone deficiency is associated with metabolic syndrome [109]. Our results suggested that a DNB analysis can capture fluctuations in gene expression towards the development of disease at ultra-early time point and the genes which are mainly expressed in the testes may be involved in triggering metabolic syndrome in adipose tissue. We also demonstrated that these fluctuations in gene expression at the pre-disease state are suppressed by the Kampo formula, a traditional Japanese medicine, Bofutsushosan extract, which is prescribed to patients with type 2 diabetes [35]. Hence, these results gave us insights into novel therapeutic strategies by targeting the pre-disease state using a combination of -omics datasets and a DNB analysis.

### 5.2. Verification of DNB Genes Using a Drosophila Model

From our previous study, we identified 147 DNB genes from the white adipose tissue of the TSOD mice and determined the pre-disease state in this mouse model [34]. Given that the pre-disease state in each disease can be determined using the DNB theory, it is necessary to determine the means by which we can reverse the physiological state from the pre-disease to healthy state, in order to make this approach clinically available in the near future. Therefore, we are developing a method called DNB intervention using the control theory, especially the sample covariance matrix, which provides a list of the genes (as we used the transcriptome dataset in this case) to be targeted, based on the simulation results of gene manipulations (either knockdown or activation). Consequently, we specified 18 genes using a DNB intervention approach from our original microarray dataset, which did not have known functions relating to metabolism. To evaluate whether these genes played a role in metabolism, we took advantage of the fruit fly model system due to its time- and cost-effectiveness compared to mouse models. We performed RNAi screening using a fat body-specific (equivalent to the adipose tissue and liver in mammals) knockdown of fly orthologs of the candidate genes to observe their resistance to starvation and found several hit genes (Akagi et al., in preparation) (Figure 3). Moreover, we confirmed that the expressions of the mouse and human genes corresponding to the fly orthologs were slightly changed in response to high-fat diet feeding in mice or human subjects with a high body mass index, from a database analysis [110]. These slight changes in expression are often disregarded or impossible to detect by using the current average-based detection of the static molecular biomarkers. Using the DNB theory, however, we can capture those “ignored genes” with a significant biological meaning. These approaches will be the model case for exploiting DNB analyses in research on aging and age-related diseases.

The advent of wearable devices or recent advances in a comprehensive monitoring platforms have made longitudinal studies using human and mammalian models much easier. However, invertebrate models such as the fruit fly (*Drosophila melanogaster*) continue to be useful as models for understanding human diseases and aging due to their short lifespan, the availability of powerful genetic manipulation tools, and the conservation of biological processes and signaling pathways between mammals [111,112]. Although fruit flies show cellular senescence phenotypes [113,114,115], flies are not the best model for studying cellular senescence in the context of aging, as most cells in adult flies are postmitotic. On the other hand, fruit flies are a useful model for studying metabolic adaptation in response to dietary interventions such as dietary restriction (DR) on aging, because the nutrient-sensing pathways, including insulin/insulin-like growth factor signaling (IIS), target of rapamycin (TOR), and the pathways involved in energy metabolism, are highly conserved across species [116,117]. DR without malnutrition is the most robust non-genetic intervention to date that can maximize the lifespan and health span in diverse organisms [116,118]. Besides extending longevity, DR has been shown to improve the tissue homeostasis, including brain [119], muscle [120], intestine [121], eyes [122], and the peripheral circadian clocks [123], in *Drosophila*. It also significantly improves metabolic health, prevents obesity, glucose intolerance/type 2 diabetes, the accumulation of senescent cells, and delays the onset of sarcopenia along with frailty in mice, rhesus monkeys, and humans [124,125].

Measuring biological age using epigenetic clocks or DNA methylation (DNAm) clocks has become a standard method for quantifying or estimating the pace of aging over the last decade [126,127,128]. As the DNAm levels of certain CpG sites are strongly correlated with chronological age and have been shown to be associated with many chronic diseases and mortality, DNAm clocks are thought to reflect biological age [129]. Notably, mice subjected to DR exhibit, on average, a 20% younger biological age than their chronological age [130]. Yet, we must take into consideration that biological age is not static, but dynamic, as it undergoes rapid fluctuations upon stress and recovery from it [131]. This finding fits the concepts of homeo-allostatic aging [132] (discussed below) and/or the effect of stress on systemic resilience [133]. Further studies are needed to decipher the mechanisms of resilience and biological aging, and we speculate that the use of a fly model is compatible for this purpose because of the ease in performing dietary manipulations, tests of stress resistance, and genetic verifications in large populations of organisms.

## 6. Homeostasis and Allostasis in Aging

Homeostasis is defined as the process that maintains a biological system in a steady state (set-point) against multiple stressors [134]. Allostasis, on the other hand, is the ability to achieve stability through a dynamic change in response to various stresses [12]. In the concept of allostasis, there is no fixed set-point parameter; rather, it is variable, the value of which results from iterative chronic stressors during aging. This kind of long-term effect of the physiologic response to stress is termed as “allostatic load” [135]. Hence, the allostatic load concept gives us an early phenotype step toward the development of diagnosable disease.

Allostatic load is measurable using an index of biomarkers known as the allostatic load index (ALI) in clinical studies. The ALI is composed of six categories: neuroendocrine function (cortisol, dehydroepiandrosterone, epinephrin, norepinephrine, dopamine, and aldosterone), immune function (IL-6, TNF-alpha, c-reactive protein, and IGF-1), metabolic function (HDL/LDL-cholesterol, triglyceride, glycosylated hemoglobin, blood glucose, insulin, albumin, creatinine, and homocysteine), cardiovascular/respiratory function (systolic blood pressure, diastolic blood pressure, peak expiratory flow, and heart rate), and anthropometric measurements (waist-to-hip ratio and body mass index) [136,137]. It is noteworthy that each allostatic mediator is interconnected in a nonlinear network, and many body systems are influenced by the same mediators [138]. Over time, a chronic allostatic load dysregulates or breaks this interconnected network, leading to disease onset, accelerated aging, and an increased mortality risk. This phenomenon is termed “allostatic overload” [139,140]. The concept of allostasis has traditionally been applied to whole-body processes, but it has recently been demonstrated that cellular allostatic load/overload is induced by chronic glucocorticoid exposure, which leads to multiple features of cellular aging, including replicative senescence, accelerated telomere shortening, and rapid aging along the epigenetic clocks in primary human fibroblasts [141].

As aging is constructed by complex and dynamic changes, Kemoun et al. proposed the concept of “homeo-allostatic aging” [132]. When an individual is robust and homeostasis is sustained (i.e., physiologically young), the individual can respond adequately to challenges (multiple stress) and the set-point (a physiological parameter) rapidly returns to its initial value. This is because the individual has enough intrinsic capacities and resilience, which is defined as the ability of the organism to respond to stress [2]. It has been shown that age-related loss of resilience is associated with the fragility of a body system. For example, the brain exhibits impaired adaptive neuroplasticity and resilience during aging, but feeding and exercise regimens result in intermittent metabolic switching from liver-derived glucose to fat-derived ketones, which facilitates this switch [142,143]. During aging, as soon as the set-point not only decreases slowly, but also does not return to its initial value after being challenged, one falls into allostasis. This is a consequence of decreases in intrinsic capacities and resilience [132]. Therefore, understanding the nature of resilience mechanisms and accumulated damage, as well as finding methods for assessing them, is critical to uncovering the underlying biological mechanisms of aging and longitudinal changes in aging trajectories [144]. In a mouse model, Chen et al. recently developed a comprehensive platform for scoring physiological aging and resilience using a multi-dimensional longitudinal phenotyping approach [145]. This platform is capable of longitudinally evaluating aging in hundreds of mice across an age range from 3 months to 3.4 years. Using the multi-dimensional data they obtained, they developed a new method for determining biological age called the Combined Age and Survival Prediction of Aging Rate (CASPAR) model, which is trained to simultaneously predict both chronological age and survival time [145]. This approach will provide novel insights into disentangling biological aging from chronological aging. Importantly, concepts arising from a dynamical systems perspective have been applied to quantify or assess the resilience in physiological systems [133,146]. Thus, we believe that the DNB theory will help to understand the mechanisms of the resilience of a body system.

## 7. Conclusions and Future Perspectives

Here, we outline the concept and usage of the DNB theory for the prediction of disease and cell fate transitions, in addition to several advantages of Raman spectroscopy, and provide potential applications for studying aging, including the prediction of senescent cell appearance (Figure 4). There are ongoing revolutions in multi-omics, artificial intelligence technologies, and bioinformatic methodologies that open up new possibilities for understanding aging science and accelerating drug discovery. Notably, a combination of modern machine learning techniques and approaches from the dynamic systems theory has successfully described the aging process in mice using large sets of longitudinal measurements [147]. Another study from the same group using longitudinal datasets of human blood samples demonstrated organism state fluctuations and an age-dependent loss of physiological resilience measured by a dynamic organism state indicator (DOSI), which comprises a single variable of neutrophil-to-lymphocyte ratio and red cell distribution width [148]. In addition, a combination of machine learning, logistic regression models, and structural equation modeling unveiled a multi-organ aging network, in which an organ’s biological age selectively influences the aging of other organs and predicts the risk of age-related morbidity and mortality [149]. Therefore, the DNB theory, which is based on the bifurcation theory of nonlinear dynamical systems, will fit in analyzing the complex trajectories of aging.

The spatial mapping of senescent cells provides critical information about their communication networks with neighboring cells as a driver of aging, but there are still some technical limitations due to the size of senescent cells, the long duration of imaging acquisition, and sample preparation and preservation [10]. In contrast, the accumulation of senescent cells in tissues may be estimated by the specific patterns of circulating proteins in the blood [150]. Notably, all of the hallmarks of aging directly or indirectly cause an inflammatory state, also known as “inflammaging”, suggesting that the pro-inflammatory state observed in many older persons may reflect the burden of biological aging [6,144,151,152]. Consistent with this notion, inflammation measured by the circulating levels of IL-6 is the only known cross-sectional and longitudinal predictor of multimorbidity and one of the strongest predictors of frailty and disability in daily life. Hence, the expression of IL-6 is proposed to serve as an early warning sign for the burden of multimorbidity [153]. As the level of IL-6 expression is included in ALI [136], analyzing ALI using the DNB theory will be fascinating to test as a novel methodology for the prediction of age-related loss of resilience, as well as the progression of cellular senescence in humans. Numerous studies have suggested that aging occurs through complex and dynamic changes [131,132,144,147,148]. This conceptual shift by “dynamical aging” opens up new and previously unexplored opportunities for the research on aging and disease prediction; thus, the DNB theory should be applied to multiple aging models with multimodal applications.

## Figures and Tables

**Figure 1 cells-12-02297-f001:**
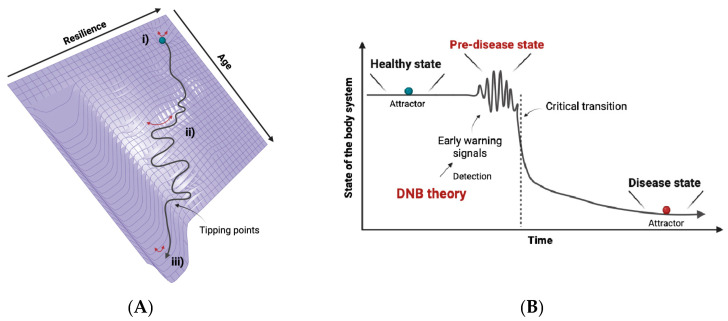
Schematic representation of aging trajectories and the DNB theory. (**A**) Representative loss of resilience along aging trajectories is drawn over the potential energy landscape. (i) Represents the homeostatic (young or healthy) state with fast and small fluctuations in the physiological parameter. (ii) Represents the allostatic state with slow and large fluctuations. (iii) Represents the allostatic overload (old or disease) state after the system crosses the tipping points and falls into a different state. This allostatic overload state is considered to be stable, and thus it is difficult to return to the original state. (**B**) The concept of the DNB theory and the pre-disease state. The DNB theory can detect early warning signals, which are sign of critical transition. The time-point just before the critical transition to the disease state (symptom onset) is named the pre-disease state.

**Figure 2 cells-12-02297-f002:**
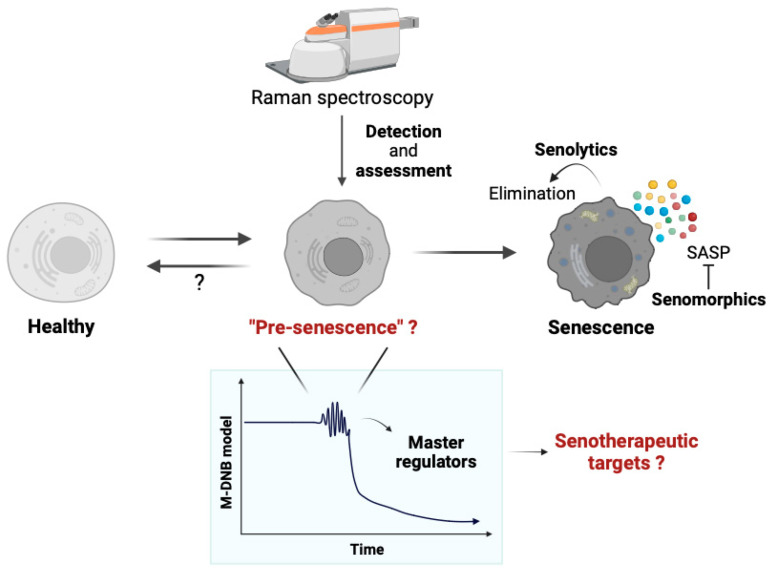
A hypothetical diagram of pre-senescence cell identification using Raman spectroscopy and the DNB theory. Given that development of senescent cells is accompanied by the cell fate transition from the proliferative to the non-proliferative state, “pre-senescence state” may exist just before the critical transition to the senescence state. If that is a case, pre-senescent cells can be identified by the DNB theory and Raman spectroscopic observation. Furthermore, M-DNB model may help to identify the master regulator genes, which commit to this critical transition. These genes may serve as novel senotherapeutic targets, in addition to the current senotherapies, including senolytics and senomorphics.

**Figure 3 cells-12-02297-f003:**
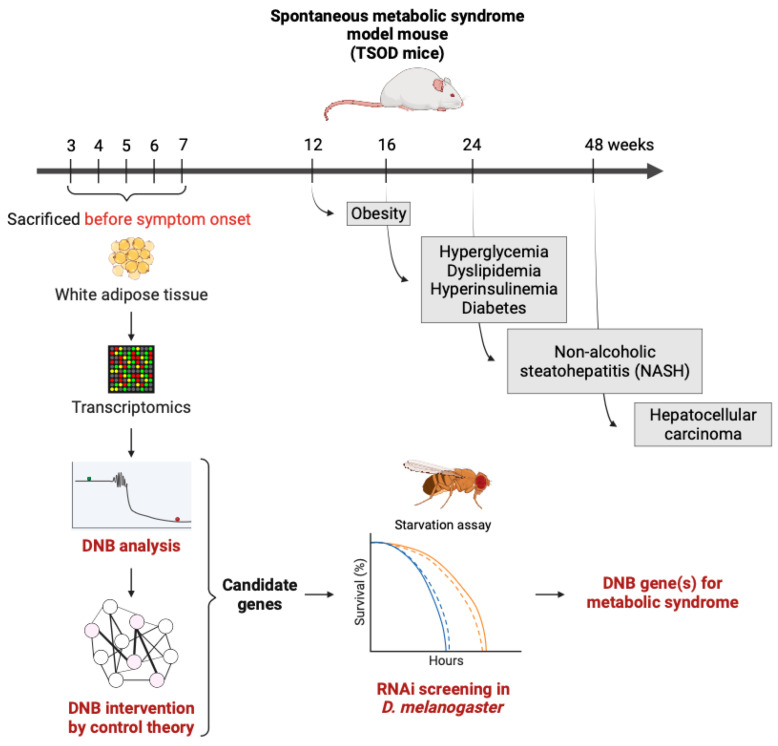
Identification and verification of the DNB genes for metabolic syndrome. The DNB analysis is capable of identifying the cluster of the genes that show large fluctuations with the strongest correlations just before the critical transition. These genes were further analyzed by the control theory, then RNAi screening was performed using the fruit fly model.

**Figure 4 cells-12-02297-f004:**
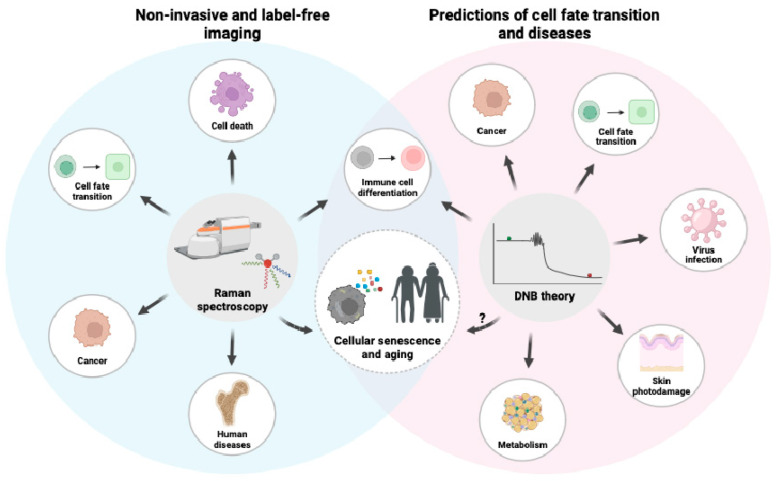
The current applications of Raman spectroscopy and the DNB theory. Non-invasive and label-free imaging using Raman spectroscopy has been used to distinguish not only senescent cells, but also various cells and human diseases. The DNB theory has been successfully predicted the cell fate transition of several cell types as well as symptom onset. A combination of Raman spectroscopy and the DNB theory may help to predict the emergence of senescent cells and age-related loss of resilience.

**Table 1 cells-12-02297-t001:** The applications of DNB theory to diseases and development.

Models	Cell Types or Species	Datasets	References
Influenza A (H3N2) infection	Human	Microarray of the blood samples	[24,26]
COVID-19 infection	Human	Case reports in five different countries and regions	[25]
Hepatocellular carcinoma	Xenograft mouse model of HCCLM3 cells	Microarray of the liver samples	[27]
Breast cancer	Human breast adenocarcinoma MCF-7 cell line	RNA-seq of MCF-7 cells	[28]
Skin photodamage	The LSE model (3D skin model consisting of normal human keratinocyte and melanocyte)	RNA-seq of the LSE model	[29]
Lung cancer	*Kras^LSL-G12D/+^; Lkb1^flox/flox^* (*KL*) mice	RNA-seq of the *KL* lung samples	[30]
Hematopoietic stem cell differentiation	Mouse hematopoietic stem cells (mHSCs)	scRNA-seq of mHSCs	[31]
Embryonic stem cell differentiation	Human embryonic stem cells (hESCs)	scRNA-seq of hESCs	[32]
Immune cell differentiation	T cells from DO11.10 TCR mice	Raman imaging	[33]
Metabolic syndrome	Metabolic syndrome model mouse (TSOD mice)	Microarray of the adipose tissues	[34,35]
Type 2 diabetes	Diabetes model rat (GK rats)	Microarray of the adipose tissues	[36]

**Table 2 cells-12-02297-t002:** Senolytic targets and agents.

Senolytic Targets	Compound	References
SRC	Dasatinib	[76]
BCL-2 family	Quercetin, Navitoclax, A1331852, A1155463, Procyanidin C1	[77,78,79]
HSP90	Geldanamycin, Tanespimycin, 17-DMAG, Ansamycin, Resorcinol	[80,81]
PI3K	Fisetin, Luteolin, Enzastauin	[77,81]
p53-FOXO4	FOXO4-DRI	[82]
Na^+^/K^+^ ATPase	Ouabain, Digoxin, Proscillaridin A, Bufalin	[83,84]
GLS1	BPTES	[85]

**Table 3 cells-12-02297-t003:** Senomorphic targets and agents.

Senomorphic Targets	Compound	References
mTOR, Nrf2, NF-κB	Rapamycin	[87,88]
NF-κB, Nrf2/GPx7, Insulin/IGF-1, mTOR etc.	Metformin	[89,90,91]
SIRT1	Resveratrol, Sirtuin-activating compounds	[92,93,94,95]
NF-κB	SR12343	[96]
p38MAPK	SB203580, UR13756, BIRB796	[97,98]
JAK/STAT	Ruxolitinib	[99,100]
ATM	KU-55933, KU-60019	[101,102]
HMG-CoA reductase	Atorvastatin, Pravastatin, Pitavastatin, Simvastatin	[103,104]
IRAK1/IκBα/NF-kB	Apigenin, Kaempferol	[105]

## Data Availability

Not applicable.

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
