# Peer review of "New Possibilities for Evaluating the Development of Age-Related Pathologies Using the Dynamical Network Biomarkers Theory"

_cells, 2023, doi:10.3390/cells12182297_

Round 1

Reviewer 1 Report

This review article cells-2593877

is very interesting and contains bright ideas, recent findings, and novelties.

The described system biological approach to the functioning of human organisms or any life system based on non-linear thermodynamics is very progressive and the methods suggested and employed, such as the DNB (dynamic network biomarkers tested in single cells transcriptomes and also the comparable Raman spectroscopy) are potentially capable to provide the dynamic integrative information on the state of the organism and the onset of disease. The notion of resilience load or a threshold of adaptation to stress for evaluating the health and ageing of an organism as a whole are very useful, indeed!

The authors use the detection of the strong oscillatory behaviour of gene clusters before the bifurcation of cell fates as the tipping points to predict the diseases associated with aging presuming in general the shift between health and disease. Theoretically and in several cases practically, it is likely OK and can be helpful in clinics.

But I realise serious misunderstandings in the biological aspects: From the title one can guess that cellular senescence is part of ageing and the widely used formulation is given in the text:

 Cellular senescence is the irreversible arrest of cell proliferation that occurs when cells are exposed to a variety of stressors, such as genotoxic agents, nutrient deprivation, hypoxia, mitochondrial dysfunction, or oncogene activation (lines 43-44).

 This definition stems from the famous article of Braude and Roninson (2001) “Īf not apoptosis then what?” and posits the irreversibility of cellular senescence. This paradigm was exploited by countless studies, consortia, programs, etc – applying  senolytics and similar with limited success as indicated by the authors.

BUT:

1 The authors have a narrow sense of what are senescing cells.

2. The authors do not formulate cancer which appears only in the scheme on Fig.4 from which it is presumed that cancer is different from cellular senescence.

Both points are mutually doubtful.

The latest data indicate that cellular senescence may be reversible and in this way, it is a gate (another face) to rejuvenation, reprogramming and stemness and thus part of cancer (carcinogenesis and in resistance to genotoxic treatments, as well).

 Moreover,  cellular senescence  is really characterised by oscillations between proliferation-stimulating and inhibiting regulators but the exit may be the bifurcation between death and resistant cancer (notably, in a very very small proportion of cells) but not just between proliferation and its stop, because cancer-driving polyploidy is inserted through mitotic slippage

 We also studied the senescence of IMR90- cells and found that a proportion of cells tetraploidise,  gain DNA breaks, p16 and NANOG-positivity in the same pre-senescent cells (so, these may be revertants from proliferative senescence). Only terminally senescent cells swell (Huna et al., 2011 https://www.ncbi.nlm.nih.gov/pmc/articles/PMC4825594/pdf/kccy-14-18-1056948.pdf

 As well, we found several-day-long fluctuations of (OCT4/p21) oscillator in breast cancer cells which finally recover from doxorubicin treatment http://www.landesbioscience.com/journals/cc/article/23285/?show_full_text=true&

Please, also see reviews https://www.intechopen.com/books/senescence-physiology-or-pathology/accelerated-senescence-of-cancer-stem-cells-a-failure-to-thrive-or-a-route-to-survival-

and  Paradoxes of cancer: survival at the brink. Semin Cancer Biol 2022 Jun; 81:119-131doi: 10.1016/j.semcancer.2020.12.009, as well as several articles by other authors of this issue devoted, in particular also to the current understanding of cellular senescence:  Jinsong Liu, Jekaterina Erenpreisa, Ewa Sikora. Polyploid giant cancer cells: An emerging new field of cancer biology, Seminars in Cancer Biology; 10.1016/j.semcancer.2021.10.006

As to the second point, cancer may be defined by a gene network, I agree, maybe it can be monitored through DNB and Raman but, please, tell the markers and tell the thresholds, if you know them!

It is also worth mentioning that the incidence of cancer in the 21st century is accelerating and half of the population in Western countries will age with and die from it. doi:10.1001/jamaoncol.2021.6987).

All methods have their advantages and restrictions. The latter should be discussed, more in detail.

For that purpose in Raman spectroscopy, the recent review can be used. https://www.frontiersin.org/articles/10.3389/fbioe.2022.856591/full

 Minor: The authors report their own studies on genetically obese animals that found impaired reproduction such as spermatogenesis and spermatid development that have never been reported as genes linked with metabolic syndrome.

In fact, testosterone deficiency in obesity is well-known to endocrinologists and andrologists. It is associated with the conversion of steroid hormones in body fat by aromatase (testosterone into oestrogen) and also the hypothalamus-gonad axis. Zitzmann, M. Testosterone deficiency, insulin resistance and the metabolic syndrome. Nat Rev Endocrinol 5, 673–681 (2009).

The quality of English is good.

Reviewer 2 Report

In the present paper, the authors review two approaches that would help assess and develop potential applications for the study on aging. Specifically, they address the development of cellular senescence dynamical network biomarkers (DNB) theory, which allows for the prediction of state transition in living organisms and Raman spectroscopy, as a non-invasive and label-free imaging technique.

The figures in the manuscript are informative and the paper is well written; in my opinion the topic will be of interest to researchers in the field. I do, however, have a few suggestions that would benefit the paper:

  1. Line 85: the authors mention the word “Mebyo” an mention it roughly means a pre-disease state. Is it possible to translate the word directly in English? It would be interesting for the readership to learn about the direct translation.

  1. When discussing the “early-warning signals, the authors state that: “Critical slowing down exhibits increases in both variation of fluctuation and autocorrelation as well as slow recovery from perturbation just before the critical transition.” The meaning of this statement is relatively straightforward. However, it would be great if the authors could provide an example of such variation of fluctuation – this would help the reader to relate with the topic even better.

  1. Can you describe in more detail the ways to measure senescence with Raman spectroscopy (a few examples would again be beneficial).

  1. There are two typos: 1) line 158 (the word “is” is missing) and 2) line 378 (delete the word “been”).

Minor editing of English language required.

Round 2

Reviewer 1 Report

The quality of English is good.
